# The Expression Regulation and Cancer-Promoting Roles of RACGAP1

**DOI:** 10.3390/biom15010003

**Published:** 2024-12-24

**Authors:** Jiacheng Lin, Yuhao Zhu, Zhaoping Lin, Jindong Yu, Xiaobing Lin, Weiyuan Lai, Beibei Tong, Liyan Xu, Enmin Li, Lin Long

**Affiliations:** 1Department of Biochemistry and Molecular Biology, Shantou University Medical College, Shantou 515041, China; 2The Key Laboratory of Molecular Biology for High Cancer Incidence Coastal Chaoshan Area, Shantou University Medical College, Shantou 515041, China; 3Institute of Oncologic Pathology, Shantou University Medical College, Shantou 515041, China

**Keywords:** RACGAP1, Rho-GTPases, ECT2, STAT3

## Abstract

RACGAP1 is a Rho-GTPase-activating protein originally discovered in male germ cells to inactivate Rac, RhoA and Cdc42 from the GTP-bound form to the GDP-bound form. GAP has traditionally been known as a tumor suppressor. However, studies increasingly suggest that overexpressed RACGAP1 activates Rac and RhoA in multiple cancers to mediate downstream oncogene overexpression by assisting in the nuclear translocation of signaling molecules and to promote cytokinesis by regulating the cytoskeleton or serving as a component of the central spindle. Contradictorily, it was also reported that RACGAP1 in gastric cancer could inactivate Rac and RhoA. In addition, studies have revealed that RACGAP1 can be a biomarker for prognosis, and its role in reducing doxorubicin sensitivity poses difficulties for treatment, while the current drug targets mainly focus on its downstream molecule. This article mainly reviews the expression regulation of RACGAP1 and its cancer-promoting functions through oncogene expression mediation and Rho-GTPase activation.

## 1. Introduction

Rac-GTPase-activating protein (RACGAP1), also known as male germ cell RacGAP (MgcRacGAP), was first discovered in male germ cells as a human chimerin-like protein with GTPase-activating ability toward Rac, Cdc42 and RhoA, and shares structural similarity with Drosophila Rotund RacGAP (Drosophila RnRacGAP), the latter of which is a protein expressed specifically in the spermatocytes of the fruit fly. This structural similarity indicates its evolutionary conservation [1]. As downstream proteins of RACGAP1, Rac, Cdc42 and RhoA are all Rho-GTPases, a subfamily belonging to the Ras superfamily that consists of small G proteins. These Rho-GTPases generally have the following two forms: the active form, binding with GTP, and the inactive form, binding with GDP. In combination with the GTP-bound Rho-GTPases, RACGAP1 can turn GTP into GDP via hydrolysis. Thereby, RACGAP1 can inactivate Rho-GTPases. Furthermore, the GTP-loading state of Rho-GTPases plays its role through the interaction with downstream effector proteins, most of which act as kinases or scaffolds to affect the cell cycle [2]. Therefore, as a GTPase-activating protein, RACGAP1 seems to function indirectly through the regulation of Rho-GTPase activity.

However, there is a controversy in current studies concerning the GTPase-activating protein (GAP) activity of RACGAP1 toward Rho-GTPases. Inconsistent with the previously reported potential inactivation of Rho-GTPases, RACGAP1 was also found to promote RhoA activation [3]. Moreover, studies have revealed that RACGAP1 does not always play its GTPase activity role in regulating cell behaviors when interacting with GTP-bound Rho-GTPases. Instead, it can also be combined with GTP-bound Rac1 without hydrolyzing GTP until having transported STAT3 into the nucleus, which means that RACGAP1-mediated nuclear translocation is triggered in a GTPase-activation-independent manner [4].

The functions of RACGAP1 are associated with cancers. RACGAP1 has been reported to overexpress in multiple cancers, participate in the development of cancers and induce drug resistance [5,6]. The role of GAP was previously considered to antagonize the role of the guanine nucleotide exchange factor (GEF), which exchanges the GDP of Rho-GTPases with GTP, but it appears that RACGAP1 plays its role collectively with GEF rather than against GEF [2]. To our knowledge, there has been no review focusing on the regulation of RACGAP1 and its roles. Therefore, in this review, we manage to elucidate the expression regulation of RACGAP1 in cancers, including its upstream regulation and feedback regulation. We also describe its cancer-promoting mechanisms from the perspective of the regulation of transcriptional factors, such as STAT3, and the influence on actin accumulation and mitosis via Rho-GTPases and GEF.

## 2. RACGAP1 Overexpression in Multiple Visceral Cancers

Current studies support that *RACGAP1* acts as an oncogene in multiple viscera, including the digestive system, respiratory system, urinary system and genital system. Most of the studies are based on bioinformatics, and the molecular mechanisms are not clear.

In the digestive system, RACGAP1 is overexpressed in both the digestive tract and digestive glands, especially in hepatocellular carcinoma (HCC) for digestive glands. It was revealed that RACGAP1 may be involved in immune cell infiltration and act as an independent prognostic factor for HCC [7]. Clinically, cirrhosis is one of the risk factors of HCC, the molecular mechanism of which may involve RACGAP1, as the recurrence-free survival and overall survival of the disease progression can take RACGAP1 as one of the independent prognostic factors [8,9,10]. In association with cell invasion and migration, as well as the early recurrence of HCC, the overexpressed RACGAP1 interacts with the polo-like kinases that function in mitosis [11]. The application of weighted gene co-expression network analysis shows a possible influence of RACGAP1 expression on the pathological T stage, histologic grade and cellular proliferation of HCC, implying that RACGAP1 can be a prognostic factor or treatment target for HCC [12]. The early discovery of HCC can take RACAP1 as a biomarker [13]. In gallbladder cancer, RACGAP1 is upregulated in targets like DNA ligase 3 (LIG3) to mediate cell growth promotion and apoptosis reduction [14,15]. In pancreatic cancer, RACGAP1 can serve as a telomere-related biomarker for prognosis [16].

For the digestive system, the expression of RACGAP1 is also upregulated. The development of gastric cancer involves RACGAP1 in the interference of stemness properties, enabling RACGAP1 to be a possible treatment target for gastric cancer [17]. By using immunohistochemistry, it was indicated that, at the invasive front, RACGAP1 expression correlates with the exacerbation of gastric cancer [18]. RACGAP1 may reduce cell autophagy and apoptosis through SIRT1/Mfn2 regulation [19]. Intriguingly, it is notable that RACGAP1 expresses at a lower level in gastric cancers than in normal tissues, which is inconsistent with most other studies [20]. Cancers in other segments of the digestive tract are also associated with RACGAP1. It is indicated that RACGAP1 is overexpressed in esophageal cancer to be a prognostic factor for poor survival and the TNM stage [21]. It is also revealed that RACGAP1 expression is associated with malignant biological behaviors of colorectal cancer cells and with the progression of colorectal cancer into a poorer T stage, invasion and lymph node metastasis, as well as recurrence [22,23]. These results support that RACGAP1 participates in cancers of the esophagus, stomach and large intestine.

In the respiratory system, RACGAP1 can be one of the overexpressed genes to identify malignant tissues from the nonmalignant tissues of lung adenocarcinomas (LUADs) with a high specificity and sensitivity, and its expression also significantly increases at the protein level in association with the poor survival [24,25]. RACGAP1 may have an effect on tumor stemness in LUADs and promote proliferation and migration [26]. The progression-free survival of nasopharyngeal carcinoma is related with RACGAP1 overexpression [27].

In the urinary system, RACGAP1 is upregulated in bladder cancer. With radical cystectomy for patients as the source of urothelial carcinoma of the bladder, RACGAP1 shows a correlation with the tumor size in support of its role as a prognostic factor for overall survival [28]. In addition, RACGAP1 belongs to a set of overexpressed hub genes associated with cell proliferation in bladder cancer, especially in the basal subtype that displays a higher expression and the highest stem cell characteristics of these genes, all of which may have AURKB and PLK1 as their upstream genes [29]. Furthermore, one study on superficial bladder transitional cell carcinoma suggests that the genes involved in cell cycle regulation and proliferation signaling are overexpressed, among which RACGAP1 together with PCNA and Hmmr may be significant prognostic biomarkers in the early development of the cancer [30]. In addition to bladder cancer, bioinformatics and machine learning also imply RACGAP1 involvement in the development of lung carcinoma-associated membranous nephropathy (MN) [31]. Together, these results imply that RACGAP1 expresses abnormally in various types of bladder cancers and probably in some other diseases of the urinary system.

In the genital system, RACGAP1 overexpression is detected, particularly in breast cancer. RACGAP1 is found to be upregulated and highly related to the clinical TNM stages of breast cancer [32,33]. Based on the data from The Cancer Genome Atlas (TCGA), RACGAP1 is one of the genes related to breast cancer prognosis [34]. In cervical cancer, RACGAP1 is considered to be associated with the cell cycle and proliferation via CDC25C [35]. The pathophysiological process of ovarian cancer possibly involves RACGAP1 [36,37]. A prognostic model for prostate cancer takes RACGAP1 as one of its five key lactylation-related genes [38]. These indicate that RACGAP1 is associated with the progression of genital system-related cancer.

Although RACGAP1 is found to be correlated with various molecules in cancers, these studies rely on bioinformatics, and more experiments are needed to elucidate how RACGAP1 promotes cancer development.

## 3. Expression Regulation of RACGAP1

Current studies indicate that RACGAP1 is generally upregulated by the aberrant expression of non-coding RNAs and transcriptional factors in various cancers. Meanwhile, a positive feedback loop is formed in some cancers to promote RACGAP1 expression.

### 3.1. Upregulation of RACGAP1 in Cancers

Multiple studies reveal that non-coding RNAs play a significant role in the expression regulation of RACGAP1. In pancreatic ductal adenocarcinoma and breast cancer, miR-204-5p was downregulated, while RACGAP1, as the target of miR-204-5p, was upregulated [39,40]. Another article indicated that MAGI2 antisense RNA 3 (MAGI2-AS3), a long non-coding RNA (lncRNA), acted as a tumor suppressor via the recruitment of lysine-specific demethylase 1A (LSD1; KDM1A) to the promoter of RACGAP1 and to the demethylase histone H3K4me2, so that the transcription of RACGAP1 was inhibited [41]. RACGAP1P, a pseudogene, was reported to be upregulated in hepatocellular carcinoma, leading to the overexpression of RACGAP1 by competitively binding with miR-15-5p, the latter of which could inhibit RACGAP1 expression [42]. Similarly, hsa_circ_0088364 and hsa_circ_0090049, which are upregulated in hepatocellular carcinoma, are associated with the upregulation of RACGAP1 at the mRNA level [43]. In colorectal cancer, hsa_circ_0001955, hsa_circ_0071681 and hsa_circ_000240 could affect survival via RACGAP1 regulation [44,45]. In ovarian cancer, circASXL1 inhibited the miR-320d-mediated RACGAP1 downregulation [46]. Moreover, lncRNA prostate androgen-regulated transcript 1 (PART1) targeted miR-6884-5p to regulate RACGAP1 expression [25]. Taken together, miRNAs and lncRNAs inhibit the expression of RACGAP1 at the translational level and transcriptional level, respectively, and their downregulation leads to the upregulation of RACGAP1 in cancers, while ceRNAs and circRNAs promote the expression of RACGAP1 by competitively binding with miRNA.

In addition to non-coding RNAs, transcriptional factors also participate in RACGAP1 expression regulation. The E2F family can cause the overexpression of RACGAP1 in squamous cell carcinoma. As one of its members, E2F3 leads to RACGAP1 overexpression in association with poor overall survival in esophageal squamous cell carcinoma and cell cycle dysregulation [47,48]. Consistently, RACGAP1 induced doxorubicin resistance in squamous cell carcinoma, resulting from the upregulated E2F7 binding to the RACGAP1 promoter [49]. E2F1-mediated RACGAP1 upregulation seemed to cause the progression of neuroendocrine prostate cancer featured with the lack of androgen receptor signaling [50]. The transcription of RACGAP1 is also upregulated by YY1 in glioma [51], Recombinant GA-Binding Protein Transcription Factor Alpha (GABPA) in hepatocellular carcinoma [52], FOXM1 in cervical cancer [53] and rbp-Jkappa in prostate cancers [54]. In conclusion, RACGAP1 overexpression can be caused by various transcriptional factors, especially the E2F family.

### 3.2. Feedback Regulation of RACGAP1

Current studies reveal that feedback regulations are common in RACGAP1 expression. It was elucidated that the miR4324-RACGAP1-STAT3-ESR1 loop led to the positive feedback of RACGAP1 overexpression in bladder expression [6]. Furthermore, RACGAP1 might mediate STAT3 phosphorylation and nuclear translocation, and, in turn, nuclear p-STAT3 could regulate RACGAP1 transcription [3,4,6]. Similarly, RACGAP1 and Sphk1 promote the expression of each other transcriptionally [55]. Moreover, the expressions of AURKA and RACGAP1 are positively correlated [20]. In addition, a study on prostate cancer also reported the reciprocal promotion between RACGAP1 and androgen receptors (ARs) [35]. HIF-α and RACGAP1 seemed to promote the expression of each other in HCC [56]. To date, the feedback loops discovered are usually positive feedbacks, which is opposite to the general understandings that negative feedback should be prevalent phenomena to maintain a balance. This may suggest that the lack of negative feedback regulation is one of the reasons for the overexpression of RACGAP1 in many cancers.

However, many studies only displayed their findings that RACGAP1 might be correlated with other proteins, but most of the molecular mechanisms remain to be elucidated. STAT3 is one of the molecules whose interaction with RACGAP1 has been elucidated in detail.

According to current studies, STAT3 is significant in the positive regulation of RACGAP. The overexpression and adverse function of RACGAP1 in bladder cancer has undergone confirmation; the underlying mechanism possibly relies on the downregulation of miR-4324 and ESR1, and the upregulation of RACGAP1 and p-STAT3 in the miR-4324-RACGAP1-STAT3-ESR1 feedback loop [6]. In this study, it was confirmed that miR-4324 downregulation may lead to RACGAP1 upregulation, the latter of which can increase STAT3 phosphorylation. As a result, p-STAT3 can bind DNA methylation transferase 3B (DNMT3B) to assist in ESR1 promoter methylation. The methylation of ESR1, whose expression production is a transcriptional factor of miR-4324, inhibits the expression of miR-4324, and the lack of miR-4324 leads to RACGAP1 overexpression, as mentioned above, to form a regulatory loop. As a supplement to these results, it was discovered that STAT3 can bind the RACGAP1 promoter to upregulate its transcription in multiple cancers [3]. Therefore, apart from the miR-4324-RACGAP1-STAT3-ESR1 feedback loop, RACGAP1 and STAT3 may directly form a feedback loop, in which RACGAP1 promotes STAT3 phosphorylation and p-STAT3 translocation enhances RACGAP1 expression. In view of this result, STAT3 is significant in this positive feedback loop.

Multiple studies have discovered the relationship between RACGAP1 and STAT3 phosphorylation but do not explore the underlying mechanism. Kawashima et al. have made contributions to unraveling the interaction between RACGAP1 and STAT3 [4,57]. It was implied that the formation of the RACGAP-Rac1-STAT3 complex might be significant for STAT3 phosphorylation. They observed the interesting phenomenon that RACGAP1 knockout reduces STAT3 phosphorylation, while RACGAP1 addition does not seem to increase STAT phosphorylation. A hypothesis that they proposed is that RACGAP1 may bind STAT3 to transport it to JAK2, a kinase localized on the cell membrane that can phosphorylate STAT3. Consistently, in the experiments of Ge et al. (2019), RACGAP1 restoration after knockdown by miR-4324 increased STAT3 phosphorylation, but the role of RACGAP1 overexpression on STAT3 seemed not to be shown [6]. In addition, although Mi et al. (2016) also confirmed that RACGAP1 could phosphorylate STAT3, only RACGAP1 knockdown experiments were shown in their article, but RACGAP1 overexpression experiments were not [58]. Therefore, further experiments are needed to clarify the relation between STAT3 phosphorylation and RACGAP1 expression.

## 4. Cancer-Promoting Functions

Current studies support that RACGAP1 promotes cancers by mediating its downstream oncogenic gene expression in signal pathways, and activating Rho-GTPases instead of inactivating Rho-GTPases, despite being named a Rho-GTPase-activating protein.

### 4.1. RACGAP1 Promotes Cancers by Regulating Oncogenic Gene Expression in Signal Pathways

RACGAP1 usually promotes the development of cancers in signal pathways. In cervical cancer, it was exhibited that RACGAP1 could mediate c-Jun expression and phosphorylation [59]. In the Hippo pathway, RACGAP1 showed colocalization with the translocated promoter region (TPR) and aurora kinase B during cytokinesis in hepatocellular carcinoma. In addition, by dephosphorylation, RACGAP1 could inactivate large tumor suppressor (LATS) 1/2 and activate YAP [3] (Figure 1). ERK-DRP1 pathway activation could be triggered by RACGAP1-mediated ECT2 recruitment [60]. RACGAP1 overexpression possibly led to PI3K/AKT pathway activation and subsequently cell growth in lung adenocarcinoma [61]. Among these, the results concerning the STAT3 pathway and the Hippo pathway are controversial.

#### 4.1.1. RACGAP1 Promotes TPR Expression by Activating YAP

The Hippo signaling pathway consists of Mammalian STE20-like protein kinase (MST) 1/2, large tumor suppressor (LATS) 1/2 and YAP, in that order. MST1/2 can activate LATS1/2 via phosphorylation, while LAST can inactivate YAP and WW domain-containing transcription regulator 1 (WWTR/TAZ) via phosphorylation [62,63].

In hepatocellular carcinoma, it has been shown that RACGAP1 has an inhibitory effect on the Hippo signaling pathway to promote the development of cancer. RACGAP1 can mediate YAP activation and contribute to immunosuppression and tumorigenesis [64]. Consistently, it has been established that cisplatin treatment can enhance the expression of RACGAP1 and its nuclear translocation, as well as YAP activation, after RACGAP1 overexpression [65]. In turn, YAP can enhance RACGAP1 phosphorylation and promote its localization at the central spindle [3]. The role of RACGAP1 in the Hippo pathway also relies on RhoA and actin. RACGAP1 could induce RhoA activation and therefore mediate YAP1/TAZ activation [66]. By the activation of RhoA, RACGAP1 can enhance the accumulation of F-actin in cytoplasma [3], while previous findings reveal that, being a protein antagonizing RhoA, Rac1 can promote actin accumulation in membrane ruffles [67]. Although it has been proven to promote hepatocellular carcinoma, RhoA is considered to be a tumor suppressor in some cancers [68]. Moreover, recent studies support that APPL2 inhibits the GAP activity of RACGAP1 to upregulate the activity of Rac1 in pancreatic β-cells to promote F-actin remodeling [69]. Therefore, the underlying mechanism, that RACGAP1 activates actin and YAP, may be associated with RhoA or Rac1 in different tissues (Figure 1).

It has also been established that the activation of YAP by RACGAP1 contributes to cytokinesis through the formation of the central spindle in combination with the translocated promoter region (TPR) [3]. As a target gene of YAP, the expression of TPR was found to be enhanced by binding with YAP and TEAD at its gene promoter. TPR plays a significant role in mitosis. At the prophase of mitosis, TPR can disassemble from the nuclear pore complexes [70]; at the metaphase of mitosis, TPR interacts with RACGAP1 under spatial and temporal control in association with spindle microtubules [3]; during the metaphase–anaphase transition, TPR participates in the regulation of the spindle-assembly checkpoint and the segregation of chromosomes [71]; at the anaphase of mitosis, TPR assists in the complex formation of TPR, RACGAP1 and AURKB at the central spindle, in which TPR acts as a scaffold for AURKB to access RACGAP1 to phosphorylate RACGAP1 [3]. As reported in the early literature, RACGAP1 is an essential protein in the central spindle complex [72]. To conclude, the suppression of the Hippo pathway enhances the transcription of TPR to promote the phosphorylation of RACGAP1, and then the assembly of the central spindle containing TPR, AURAKB and RACGAP1 (Figure 1).

#### 4.1.2. RACGAP1 Promotes Survivin Expression by Phosphorylating STAT3

As mentioned above, RACGAP1 participates in STAT3 phosphorylation to regulate RACGAP1 expression and cause an oncogenic effect. Studies also show that RACGAP1 promotes the nuclear translocation of p-STAT3. It has been proposed that the RACGAP1-Rac1-STAT3 ternary complex is formed to promote STAT3 phosphorylation, followed by STAT3 nuclear translocation [4]. Based on previous studies, the protein transport from cytoplasma to the nucleus requires a special functional sequence called the nuclear localization signal (NLS). The types of NLSs can be polybasic NLSs, including monopartite or bipartite NLSs, which interact with importin α/β heterodimers, or tyrosine-containing NLSs (PY-NLSs), which directly bind importin β [73,74,75]. For RACGAP1, it has been confirmed that RACGAP1 contains bipartite NLS, and its nuclear translocation requires the participation of GTP-bound Rac1 and STAT3 to form a ternary complex [4]. Rac1 and p-STAT3 activate the NLS RACGAP1 so that it can promote the nuclear translocation and the transcriptional activity of p-STAT3 [4]. In addition, the GTP-bound Rac1 of the complex is necessary to transport STAT3 [4], suggesting that RACGAP1 cannot inactivate Rac1 via its GAP domain. In uterine carcinosarcoma, after RACGAP1 overexpression, STAT3 is phosphorylated and promotes the transcription of survivin to improve the invasive and migratory ability of cells [58] (Figure 1). Taken together, RACGAP1 may promote STAT3 phosphorylation, nuclear translocation and the expression of STAT3 downstream molecules, such as survivin, to lead to the development of cancers.

### 4.2. RACGAP1 Promotes Cancers by Rho-GTPase Activation

Previously, RACGAP1 was considered to inactivate Rho-GTPases. RACGAP1 was originally discovered to have a GTPase-activating ability toward Rac and Cdc42 but not obviously toward RhoA [1]. Its GTPase-activating ability, however, led to the inactivation of Rac, Cdc42 and RhoA by converting them from the GTP-binding state to the GDP-binding state, that is, from the active form to the inactive form [2] (Figure 2a).

However, recent studies support that RACGAP1 can activate Rho-GTPases in cancers. In studies showing RACGAP1 overexpression in squamous cell carcinoma, it is indicated that RACGAP1 can activate Rac1, but inactivate RhoA, which results from the phosphorylation of RACGAP1 at Ser387 by aurora B kinase, to convert the RACGAP1 substrate preference from Rac to RhoA, although evidence of the activation of aurora B kinase is not given [76] (Figure 2b). However, it is also reported that Rac1 inactivation caused by RACGAP1 can promote bladder cancer progression, in which the RACGAP1-SHCBP1 interaction is induced by EGF [77]. RACGAP1P, as a competitive endogenous RNA (ceRNA), is reported to be upregulated in hepatocellular carcinoma, resulting in the activation of the RACGAP1/RhoA/ERK pathway by the inhibition of miR-15-5p interference on RACGAP1 expression. For the role of RACGAP1 on RhoA, it is also implied that RACGAP1 can activate RhoA rather than inactivate it [42] (Figure 2c). Studies of the RACGAP1-mediated translocation of STAT3, which is known to participate in tumorigenesis, report that GTP-bound Rac1 in complex with RACGAP1 and STAT3 is necessary to transport STAT3, implying that RACGAP1 can keep Rac1 activated in cytoplasm [3,4,6]. Moreover, in hepatocellular carcinoma, it is indicated that RACGAP1 can activate RhoA to function, and it was verified that RACGAP1 enhances the stability of ECT2 to improve ECT2-mediated RhoA activation [3] (Figure 2d). The function of RACGAP1 in C. elegans spermatheca can be independent of its GAP activity [78]. Since many articles reveal that RACGAP1 functions in a GAP-activity-independent way, it seems that the GAP domain activity of RACGAP1 is not exhibited in the development of cancers. RACGAP1 plays its role through Rho-GTPase activation.

The family members of Rho-GTPases—Rac, Rho and Cdc42—have an influence on the activity of one another. To make it simple, a model was proposed in NIH3T3 cell lines that Rac could inactivate Rho unidirectionally, and Cdc42 could inactivate Rho indirectly via Rac activation [79] (Figure 2e). Rac was proven to participate in cell–cell contacts, actin accumulation in membrane ruffles and epithelial-like morphologies, while Rho was proven to participate in the formation of stress fibers [79]. However, the impacts of Rho-GTPases on one another still remain controversial. The model above has two points to be noticed. First, Rac can inactivate RhoA. Second, RhoA cannot, in turn, affect Rac. For the first point, it was implied that Rac could activate Rho in Swiss 3T3 fibroblasts [67] (Figure 2f). This might suggest that the activities of Rac and Rho are dependent of their contexts and not simply negatively correlated. For the second point, studies reveal that RhoA is able to change the activity of Rac. From the perspective of the functions of Rac and RhoA, recent work indicates that RACGAP1 could enhance F-actin accumulation by RhoA activation [3]. Furthermore, it was also suggested that Rac and Rho could inhibit the activity of each other [80]. In conclusion, the interplay of Rho-GTPase activity varies in different contexts, and the underlying mechanisms remain to be clarified.

### 4.3. RACGAP1 Promotes Cancers in Coordination with GEF

GAP and GEF are known to participate in the inactivation and activation of Rho-GTPases. Rho-GTPases are considered to be activated in a GTP-bound state and inactivated in a GDP-bound state. In contrast to GAP, which can inactivate Rho-GTPases, GEF can activate Rho-GTPases by exchanging GDP for GTP [81].

Although GAP and GEF antagonize each other, ECT2 belongs to GEF, but RACGAP1 promotes RhoA activation via the interaction with ECT2, and this interplay breaks via interaction with protein phosphatase 2A-B′ (PP2A-B′).

In HCC, it has been suggested that RACGAP1 functions in the ECT2-mediated Rho activation [82]. Bioinformatics and machine learning reveal that RACGAP1 together with ECT2 may be involved in HBV-HCC development [83]. In detail, RACGAP1 should be phosphorylated at two sites by Plk1 to provide a docking site for ECT2 so that RACGAP1 can bind to the BRCT domain of ECT2. Plk1 may need the anillin actin-binding protein (ANLN) as a scaffold to function on RACGAP1 [84]. ECT2 binds to RACGAP1 to activate RhoA and Myosin II, and promotes actomyosin ring formation at the anaphase. In this process, MKLP1, another key component of the central spindle that can bind to the coiled-coil domain near the N terminus of RACGAP1, is essential for RACGAP1 to bind with the BRCT domain [85,86,87]. Furthermore, the RACGAP1-ECT2-MKLP1 complex is involved in NuMA–dynein–dynactin localization away from the equatorial membrane [88]. RACGAP1 and ECT2 also interplays during the interphase, in addition to mitosis, in which ECT2 promotes the activation of RhoA and ERK [89]. In all, although it seems that GEF antagonizes the effect of GAP, recent studies suggest that RACGAP1 can coordinate with ECT2 to promote the proliferation and migration of cancer cells.

While ECT2 and RACGAP1 can activate RhoA, PP2A-B’ can break the interaction between RACGAP1 and ECT2 to inhibit RhoA activation. PP2A-B’ is a holoenzyme that is confirmed to bind with the RACGAP1 B’-binding motif LxxIxEx after Plk1 phosphorylation on RACGAP1. PP2A-B’ binding to RACGAP1 could disrupt the interaction between RACGAP1 and ECT2 via RACGAP1 dephosphorylation [87]. Therefore, RACGAP1 participates in the balance between ECT2-mediated RhoA activation and PP2A-B’-induced RhoA inactivation (Figure 3).

## 5. Conclusions

RACGAP1 is a Rho-GTPase-activating protein that has a GAP domain to inactivate Rac, RhoA and Cdc42 by converting their GTPs to GDPs. Nevertheless, studies increasingly reveal that RACGAP1 activates Rho-GTPases rather than inactivates them, which makes it a cancer-promoting protein that is different from other GAPs, which are usually known to be tumor suppressors [81]. Although Rac is considered to antagonize RhoA, studies suggest that RACGAP1 can keep Rac or RhoA activated in different cancers, and the subsequent events of Rac and RhoA activation are not always irrelevant, implying that RACGAP1 can coordinate Rac and RhoA in some situations.

As a cancer-promoting protein, RACGAP1 is upregulated in most cancers, and the abnormal expression can result from non-coding RNAs or transcriptional factors. Several positive feedback loops of RACGAP1 expression have been proven, while, to our knowledge, negative feedbacks have not been discovered, which may account for its overexpression in cancers. One study reported that RACGAP1 exhibited a low expression level in gastric cancers, thereby reducing the inactivation of Rho-GTPases [20], which is in contrast to other studies about gastric cancers. Therefore, the expression of RACGAP1 in cancers, as well as the regulatory mechanisms, may remain to be further studied.

In conclusion, this review summarizes that RACGAP1 is a Rho-GTPase-activating protein that is upregulated in multiple cancers, and that can activate Rac and RhoA to promote cytokinesis and the nuclear translocation of some transcriptional factors (e.g., STAT3) to enhance the expression of downstream oncogenic genes.

## Figures and Tables

**Figure 1 biomolecules-15-00003-f001:**
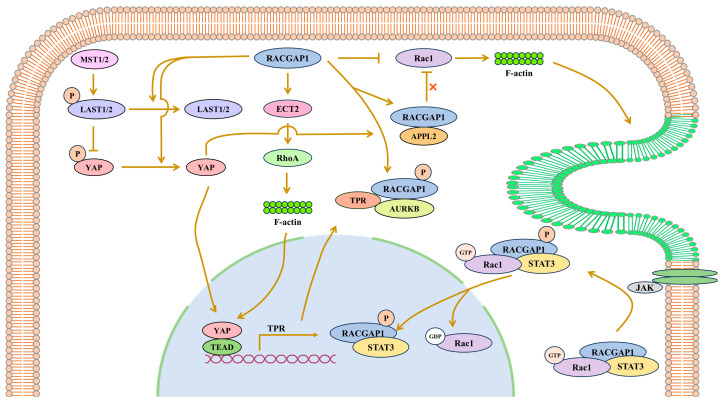
RACGAP1 promotes cancer through the Hippo pathway and JAK/STAT pathway. In the Hippo pathway, the phosphorylated form of MST1/2 activates LAST1/2 via phosphorylation. YAP can be deactivated by the activated form of LAST when phosphorylated. YAP in the non-phosphorylated state can enter the nucleus, bind to the promoter region of TPR with TEAD and promote its transcription. TPR, RACGAP1 and AURKB form complexes that promote spindle formation. Moreover, RACGAP1 inhibits the Hippo pathway by promoting the inactivation of LAST1/2 through F-actin. RACGAP1 forms a ternary complex together with STAT3 and Rac1 to phosphorylate STAT3, and accomplishes the nuclear transport of STAT3 as a nuclear chaperone. RACGAP1 binds JAK2 to enable the phosphorylation of STAT3 by JAK2. Rac1 and p-STAT3 activate the nuclear localization signal (NLS) in RACGAP1 so that it can promote the nuclear translocation and the transcriptional activity of p-STAT3. RACGAP1 can inactivate Rac1, but after binding with APPL2, it cannot inactivate Rac1.

**Figure 2 biomolecules-15-00003-f002:**
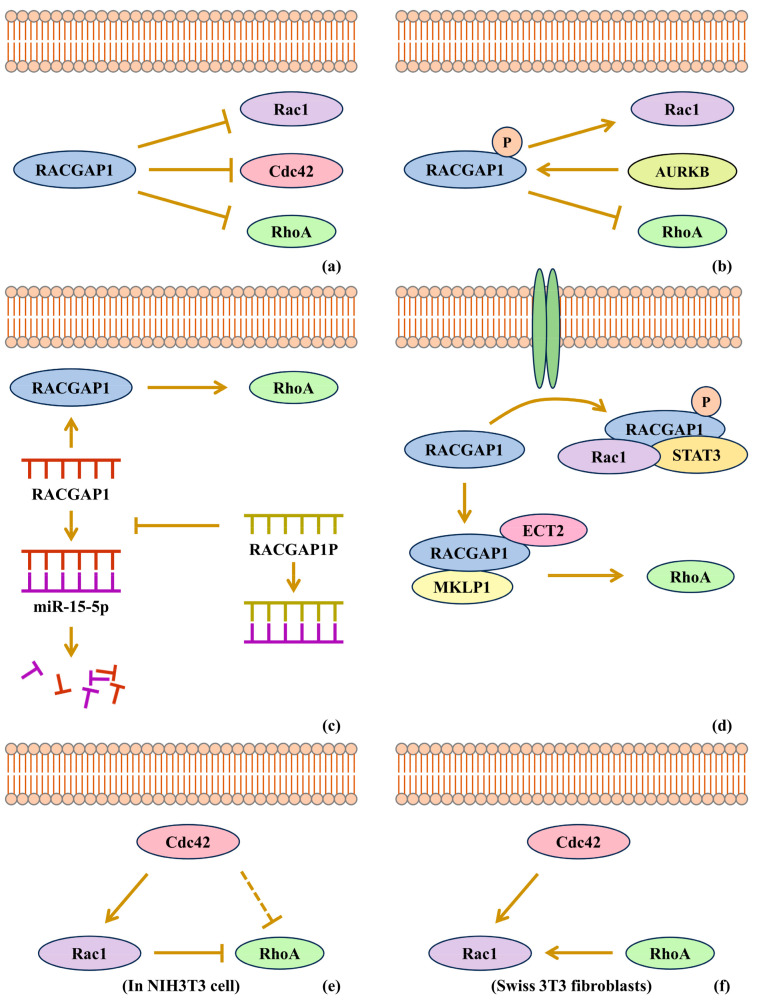
RACGAP1 can activate Rho-GTPases. (**a**) RACGAP1 inhibits Rac1, Cdc42 and RhoA. (**b**) Phosphorylated RACGAP1 can activate Rac1. (**c**) RACGAP1 can activate RhoA in hepatocellular carcinoma. (**d**) RACGAP1 can form a complex with Rac1 and STAT3, and with ECT and MKLP1, the latter complex of which can activate RhoA. (**e**) In NIH3T3 cells, Rac1 can inhibit RhoA. (**f**) In Swiss 3T3 fibroblasts, RhoA can activate Rac1.

**Figure 3 biomolecules-15-00003-f003:**
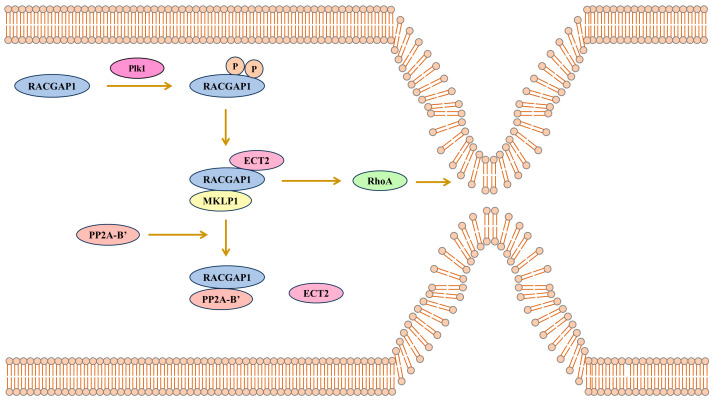
RACGAP1 binding ECT2 to promotes actomyosin ring formation. Plk phosphorylates RACGAP1 at 2 sites so that RACGAP1 binds to ECT2 with the help of MKLP1, and the complex activates RhoA, Myosin II and ERK. PP2A-B′ can bind to RACGAP1 to break the interaction between RACGAP1 and ECT2.

## Data Availability

No new data were created or analyzed in this study.

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
