# Peer review of "The Expression Regulation and Cancer-Promoting Roles of RACGAP1"

_biomolecules, 2024, doi:10.3390/biom15010003_

Round 1
Reviewer 1 Report
Comments and Suggestions for Authors
The review by Lin et al is focused on RacGAP1, an interesting but poorly understood protein. The material reviewed by the authors is extensive. The review is likely to be useful to everyone interested in the cellular functions and biological role of RacGAP1. However, several issues have to be addressed to improve this work.
1. Multiple funding are described, and the contradictions between them are presented, but no attempt has been made to present a unified picture of the RacGAP1 functions in normal and transformed cells. The reader is left with a lot of information, but the big picture remains elusive. It is imperative to present this big picture.
2. Sometimes, facts that look self-explanatory to the authors provoke some confusion. Thus, it is stated many times that RacGAP1 facilitates phosphorylation of Stat3; once it is clarified that this phosphorylation is done by a Jak kinase. What is the molecular mechanism of this RacGAP1-mediated enhancement? It is stated that RacGAP1 acts as a 'nuclear chaperone'; how? Likewise, molecular mechanisms for several other RacGAP1 functions are not elaborated. If they have been determined, they should be clearly presented. If not, it should be stated.
3. Multiple statements in the text are unclear. Here are a few examples.
line 45 the role of RACGAP1 is unnecessarily relying on its GTPase activity
line 179 RacGAP1 can indirectly phosphorylate Stat3 (and on line 196 it says that RacGAP1 can phosphorylate Stat3)
line 193 miR-4324 overexpression having been confirmed to reduce RACGAP1 did decrease the level of p-STAT3 according to the feedback loop
line 199 Since RACGAP1 was transfected by plasmid, it should locate in the cytoplasm, and to interact with plasmid, STAT3 phosphorylation to enter nucleus is unnecessary.
It is necessary to carefully re-state these and other unclear sentences.
4. Some statements are clear, but sound awkward. They have to be corrected.
line 51 RacGAP1 is distinctive
line 64 cirrhosis can exacerbate into HCC
line 325 many articles disagree with the GAP activity of RACGAP1
Comments on the Quality of English Language
In the comments
Author Response
Thank you very much for taking the time to review this manuscript. Please find the detailed responses below and the corresponding revision in the re-submitted files
Comments 1: Multiple funding are described, and the contradictions between them are presented, but no attempt has been made to present a unified picture of the RacGAP1 functions in normal and transformed cells. The reader is left with a lot of information, but the big picture remains elusive. It is imperative to present this big picture.
Response 1: Thank you for pointing this out. We agree with this comment. To make the review easy to understand, we summarized that the functions of RACGAP1 are primarily achieved by transcriptional factors like STAT3 to promote transcription and by Rho-GTPases to promote actin accumulation and mitosis from line 53 to line 57. As reader looks through the content, they can find that we mainly discuss about the two aspects from line 211 to line 373 corresponding to the end of the introduction paragraph.
In addition, we mention in line 60 and line 121 that a lot of the results are based on bioinformatics, and more evidence is needed to support those results.
Comments 2: Sometimes, facts that look self-explanatory to the authors provoke some confusion. Thus, it is stated many times that RacGAP1 facilitates phosphorylation of Stat3; once it is clarified that this phosphorylation is done by a Jak kinase. What is the molecular mechanism of this RacGAP1-mediated enhancement? It is stated that RacGAP1 acts as a 'nuclear chaperone'; how? Likewise, molecular mechanisms for several other RacGAP1 functions are not elaborated. If they have been determined, they should be clearly presented. If not, it should be stated.
Response 2: Thank you for pointing out this. We agree. We might mistakenly present the findings and cause confusion before when trying to figure out some inconsistent results in the previous studies. Therefore, we read those articles carefully again and try to express the results clearer to the reader.
From line 165 to 167, we show two functions of RACGAP1 on STAT3, that is phosphorylation and nuclear translocation.
From line 176 to line 179, we tell the reader that we will go deep into the interaction between RACGAP1 and STAT3.
From line 180 to 195, we introduce miR4324-RACGAP1-STAT3-ESR1 axis in bladder cancer in detail.
From line 196 to 210, we first point out that the previous paragraph just reveals that RACGAP1 can phosphorylate STAT3, and then show a possibly strange phenomenon that RACGAP1 knockout reduce STAT3 phosphorylation, but RACGAP1 overexpression seems not. We also point out that a hypothesis is proposed to explain this, and the mechanism still remains to be studied.
From line 275 to 279, we point out that we are going to talk about how RACGAP1 promote STAT3 nuclear translocation. Then we state that a ternary complex RACGAP1-Rac1-STAT3 is formed, then STAT3 and Rac1 can activate RACGAP1, and finally a sequence called NLS of RACGAP1 binds importin to enable the nuclear translocation of the ternary complex.
From line 291 to 293, we make the reader recall the functions of RACGAP1 on STAT3 including phosphorylation and nuclear translocation, and eventually STAT3 may promote the expression of downstream molecules to lead to cancers.
Comments 3: Multiple statements in the text are unclear.
line 45 the role of RACGAP1 is unnecessarily relying on its GTPase activity
line 179 RacGAP1 can indirectly phosphorylate Stat3 (and on line 196 it says that RacGAP1 can phosphorylate Stat3)
line 193 miR-4324 overexpression having been confirmed to reduce RACGAP1 did decrease the level of p-STAT3 according to the feedback loop
line 199 Since RACGAP1 was transfected by plasmid, it should locate in the cytoplasm, and to interact with plasmid, STAT3 phosphorylation to enter nucleus is unnecessary.
It is necessary to carefully re-state these and other unclear sentences.
Response 3: Thank you for pointing out this. We agree. Therefore, we try to rewrite the sentences to make them clear in line 45, and line 176 to 210.
Comments 4: Some statements are clear, but sound awkward. They have to be corrected.
Response 4: Thank you for pointing out this. We agree. Therefore, we try to rewrite the sentences to make them clear in line 51, line 65 and line 324.
Reviewer 2 Report
Comments and Suggestions for Authors
Dear Authors
Review describes basic mechanisms and concludes well that RACGAP1 is a Rho-GTPase activating protein that is upregulated in multiple cancers and that can activate Rac and RhoA to promote cytokinesis and nuclear translocation of some transcriptional factors (e.g. STAT3) to enhance the expression of downstream oncogenic genes.
Minor correction
1) Please keep a graphical abstract for readers to enjoy it.
Author Response
Thank you very much for taking the time to review this manuscript. We have designed a graphic abstract for readers, which focus on RACGAP1 and its role on Rho-GTPases and transcriptional factors like STAT3.
Round 2
Reviewer 1 Report
Comments and Suggestions for Authors.